# 1-Methylimidazole as an Organic Catalyst for [3+3]-Cyclodimerization of Acylethynylpyrroles to Bis(acylmethylidene)dipyrrolo[1,2-*a*:1′,2′-*d*]pyrazines

Kseniya V. Belyaeva, Lina P. Nikitina, Veronika S. Gen', Denis N. Tomilin, Lyubov N. Sobenina, Andrei V. Afonin, Ludmila A. Oparina and Boris A. Trofimov *

A.E. Favorsky Irkutsk Institute of Chemistry, Siberian Branch, Russian Academy of Sciences, 1 Favorsky Str., Irkutsk 664033, Russia
*   Correspondence: boris_trofimov@irioch.irk.ru

**Abstract:** Acylethynylpyrroles, now readily available by the cross-coupling of pyrroles with acylbromoacetylenes in solid $Al_2O_3$ media, in the presence of 1-methylimidazole underwent unprecedentedly easy (40–45 °C) cyclodimerization into bis(acylmethylidene)dipyrrolo[1,2-*a*:1′,2′-*d*]pyrazines in up to 51% yield. Some other organic and inorganic basic catalysts can also trigger this cyclodimerization, but less efficiently.

**Keywords:** alkynes; cyclodimerization; imidazole; organic catalyst; pyrazine; pyrrole

## 1. Introduction

One of the main trends in modern organic chemistry is the design of fused heterocyclic systems, particularly composed of counterparts which have pharmacologically or synthetically important properties. Such counterparts reinforce and fundamentally complement each other due to the synergetic effects. Among the fused systems of these types are pyrrolopyrazines, which merge in their molecules biologically active pyrrole [1–4] and pyrazine [5,6] structures.

It is known that functionalized pyrrolopyrazines exhibit a wide range of pharmacological activities [7], including anticancer [8], tuberculostatic [9], anti-inflammatory [10,11], antimalarial [12], antibacterial [13], anticonvulsant [14,15], etc.

The dipyrrolo[1,2-*a*:1′,2′-*d*]pyrazine scaffold is a key pharmacophoric structural motif of a number of natural and synthetic compounds (Figure 1). They include pyrocoll, a secondary metabolite found in an extract of Streptomyces sp. AK 409 (active against various strains of Arthrobacter, filamentous fungi, some pathogenic protozoa and human cancer cell lines) [16], as well as a natural alkaloid Spirotryprostatine (showing anti-mitotic properties and antitumor activity) [17,18], and metabolite of the Sorangium cellulosum strain Soce895 (producer of the antibiotic tuggacin A, Sorazinone A) [19]. Functionalized derivatives of similar tricyclic systems having common nitrogen atoms are considered as potential anticancer, antifungal [20] and antibacterial drugs. These dipyrrolopyrazines are privileged structures that can act as ligands for various biological targets and are precursors for the design of new biologically active molecules.

Moreover, in the last decade, dipyrrolo[1,2-*a*:1′,2′-*d*]pyrazines in the composition of polyconjugated indole derivatives or heteroindacenes, due to their electroluminescent properties, have found application in the composition of materials for organic electronic devices, displays [21,22], and LEDs [23].

Therefore, the development of methods for the synthesis of dipyrrolo[1,2-*a*:1′,2′-*d*]pyrazines and their functionalized derivatives is a challenge for drug-oriented chemistry and materials science.

**Figure 1.** Biologically active dipyrrolo[1,2-*a*:1',2'-*d*]pyrazines.

Analysis of literature data showed that most of the syntheses of dipyrrolo[1,2-*a*:1',2'-*d*]pyrazines are devoted to pyrocoll derivatives. In the arsenal of synthetic chemists, the number of approaches to the direct synthesis of other dipyrrolo[1,2-*a*:1',2'-*d*]pyrazines is limited to a few reactions. The reaction of pyrrolyl-2-carbaldehyde with secondary amines (three examples) led to diaminodipyrrolo[1,2-*a*:1',2'-*d*]pyrazines in up to 70% yields [24]. The ring-opening of 2-methylaminofurans under the action of acids followed by rearrangement of the intermediate gave dipyrrolopyrazines (two examples, yields were 36% and 53%) [25].

Thus, today there still remain a requirement for the development of a simple one-step synthesis of functionalized dipyrrolo[1,2-*a*:1',2'-*d*]pyrazines.

Acylethynylpyrroles **1**, now readily available by the cross-coupling of pyrroles with acylbromoacetylenes in the solid Al$_2$O$_3$ medium [26], proved to be valuable building blocks for the synthesis of pyrrolo[1,2-*a*]pyrazines in the reaction with propargylamine under the action of cesium carbonate (Scheme 1) [27].

R$^1$, R$^2$ = H, Alk,cyclo-Alk, Ar, HetAr;
R$^3$ = Ph, 2-furyl, 2-thienyl

**Scheme 1.** Synthesis of acylethynylpyrroles and Cs$_2$CO$_3$/DMSO catalyzed intermolecular cyclization with propargylamine to (acylmethylidene)pyrrolo[1,2-*a*]pyrazines.

Recently, we have found that acylethynylpyrroles **1** undergo non-catalytic exceptionally mild [3+2]-cyclization with pyrrolines to afford tetrahydrodipyrroloimidazoles in up to 93% yields (Scheme 2) [28].

**Scheme 2.** [3+2]-Cyclization of acylethynylpyrroles with 1-pyrrolines.

It seems probable that similar cyclization could take place when 1-pyrrolines are replaced by imidazoles, which also have a C=N intramolecular double bond. However, when this assumption was checked using the example of the pair benzoylethynylpyrrole **1a**/1-methylimidazole, we encountered an absolutely unexpected reaction, namely [3+3]-cyclodimerization of pyrrole **1a** into bis(acylmethylidene)dipyrrolo[1,2-*a*:1′,2′-*d*]pyrazine **2a** (36% yield), while 1-methylimidazole was completely recovered (Scheme 3).

**Scheme 3.** [3+3]-Cyclodimerization of benzoylethynylpyrrole **1a** in the presence of 1-methylimidazole.

Notably, in the absence of 1-methylimidazole, no cyclodimerization of 1a occurred, meaning that 1-methylimidazole acts as an organic catalyst. The novelty of this work is that, despite the sufficient background of studying acylethynylpyrrole reactivity with different nucleophiles [27,29], including their behavior in various basic media, similar cyclodimerization products have been observed here for the first time. This reaction, when optimized, promises a simple straightforward route to a new family of dipyrrolo [1,2-*a*:1′,2′-*d*]pyrazines decorated with $\alpha,\beta$-ethylenic keto substituents incorporated in a through-conjugated system with both pyrrole structures.

In this paper, we have shared major features of this new reaction.

## 2. Results

### 2.1. Optimization of the Reaction Conditions

A brief optimization of the reaction conditions on the example of benzoylethynylpyrrole **1a** (Scheme 4, Table 1) showed that the process proceeded with the most efficiency in the presence of equimolar amounts of 1-methylimidazole, which in this process acts not only as a catalyst, but also as a solvent (51% yield of **2a**, entry 5). Reducing the catalyst loading to 20 mol% (entry 3) or increasing it to 500 mol% (entry 6) had no positive effect on the yield of product **2a**. On the example of 20 mol% loading of 1-methylimidazole, it was found that the temperature required to start the reaction was 40–45 °C (entries 2

cf. 3). With the same catalyst loading, less polar (compared to acetonitrile) solvent 1,4-dioxane showed less efficiency (entries 4 and 3). The use of hexane as a co-solvent, together with 1-methylimidazole, reduced the time of complete ethynylpyrrole **1a** conversion to 8 h, but did not suppress the side oligomerization process (entry 7). The replacement of 1-methylimidazole (1-MeIM) with the less basic 1-phenylimidazole (1-PhIM) (pKa = 7.21 and 6.48, respectively) (entry 8) did not decrease the oligomer content and increase of dipyrrolopyrazine **2a** yield (26% yield). 1-Methylbenzimidazole did not allow the synthesis of product **2a** at all (entry 9).

**Scheme 4.** [3+3]-Cyclodimerization of benzoylethynylpyrrole **1a** in the presence of 1-substituted imidazoles.

**Table 1.** Conditions for [3+3]-cyclodimerization of benzoylethynylpyrrole **1a** under the action of imidazoles.

| Entry | Imidazoles, mol% | Solvent | Temp., °C | Time, h | Conversion of 1a, % | Yield of 2a, % |
|-------|------------------|---------|-----------|---------|---------------------|----------------|
| 1 | - | MeCN | 40–45 | 48 | ~0 | 0 |
| 2 | 1-MeIM, 20 | MeCN | 20–25 | 48 | ~0 | 0 |
| 3 | 1-MeIM, 20 | MeCN | 40–45 | 48 | 99 | 36 |
| 4 | 1-MeIM, 20 | 1,4-dioxane | 40–45 | 48 | 85 | 20 |
| 5 | 1-MeIM, 100 | - | 40–45 | 24 | 95 | 51 |
| 6 | 1-MeIM, 500 | - | 40–45 | 24 | 99 | 42 |
| 7 | 1-MeIM, 100 | *n*-hexane | 40–45 | 8 | 99 | 38 |
| 8 | 1-PhIM, 100 | - | 40–45 | 48 | 96 | 26 |
| 9 | 1-MeBIM, 100 | MeCN | 40–45 | 24 | 24 | 0 |

Then, inorganic and other common organic bases were tested as catalysts in this cyclodimerization (Scheme 5, Table 2). When KOH was used instead of 1-methylimidazole, the yield of dipyrrolopyrazine **2a** was 32% (entry 1). Changing acetonitrile for other solvents (*n*-hexane, DMSO, THF, H$_2$O) proved to be ineffective (entries 2–5). The increase in KOH loading to 50 mol% gave no improvement, since the yield of the target product **2a** decreased to 29% (entry 6). Additional screening of organic and inorganic bases showed that the cyclodimerization under the action of 20 mol% NaOH, *t*-BuOK, and DABCO ensured the yields of 20–33% (entries 7–11), while Et$_3$N, Ph$_3$P and TMEDA were ineffective at all (entries 12–14).

Thus, the optimization results (Tables 1 and 2) are evidence that the most effective catalyst was 1-methylimidazole (Table 1, entry 5).

**Scheme 5.** [3+3]-Cyclodimerization of benzoylethynylpyrrole **1a** in the presence of bases.

**Table 2.** Conditions for [3+3]-cyclodimerization of benzoylethynylpyrrole **1a** under the action of bases.

| Entry | Base, mol% | Solvent | Temp., °C | Time, h | Conversion of 1a, % | Yield of 2a, % |
|-------|-----------|---------|-----------|---------|---------------------|----------------|
| 1 | KOH, 20 | MeCN | 40–45 | 8 | 97 | 32 |
| 2 | KOH, 20 | *n*-hexane | 40–45 | 8 | 0 | 0 |
| 3 | KOH, 20 | DMSO | 40–45 | 8 | 100 | 0 |
| 4 | KOH, 20 | THF | 40–45 | 8 | 0 | 0 |
| 5 | KOH, 20 | $H_2O$ | 40–45 | 48 | 100 | trace |
| 6 | KOH, 50 | MeCN | 40–45 | 4 | 100 | 29 |
| 7 | NaOH, 20 | MeCN | 40–45 | 8 | 85 | 20 |
| 8 | NaOH, 20 | MeCN | 20–25 | 24 | 99 | 5 |
| 9 | NaOH, 20 | MeCN | 60–65 | 6 | 99 | 24 |
| 10 | *t*-BuOK, 20 | MeCN | 40–45 | 8 | 100 | 33 |
| 11 | DABCO, 20 | MeCN | 40–45 | 48 | 70 | 30 |
| 12 | Et$_3$N, 20 | MeCN | 60–65 | 48 | 0 | 0 |
| 13 | Ph$_3$P, 20 | MeCN | 70–75 | 96 | 95 | 0 |
| 14 | TMEDA, 20 | MeCN | 40–45 | 48 | 0 | 0 |

*2.2. Study on Substrate Scope*

Next, we investigated the substrate coverage in the series of acylethynylpyrroles **1** in the above cyclodimerization catalyzed by an equimolar amount of 1-methylimidazole at 40–45 °C (Scheme 6).

The heteroaromatic substituents at the carbonyl group (furyl, thienyl), seemingly additionally activating the triple bond of acetylenes **1b,c**, promoted greater oligomerization and consequently decreased the yields of cyclodimerization products **2b,c** to 23–26% vs. 51% for **2a**. Substituents in the pyrrole ring of ethynylpyrroles **1** significantly affected the reaction outcome. The substituents (Me, Ph, 4-FC$_6$H$_4$, 4-ClC$_6$H$_4$, thienyl-2) at the position 5 in the pyrrole ring of ethynylpyrroles **1d–h** diminished the yield of dipyrrolopyrazines **2d–h**, but in this case, 100% *E,E*-stereoselectivity of the reaction was achieved. Wide range of the yields (14–51%) as well as different stereoselectivity is probably due to strict steric requirements to mutual orientation of the molecules, their parts and substituents during the dimerization process. This follows particularly clearly from the fact that benzoyl- and thenoylethynyltetrahydroindole, which has a bulky pyrrole counterpart, gave the expected dimer only in 8% ($^1$H NMR) and 14% yields correspondingly for a much longer time (96 h and 144 h), while with benzoylethynylpyrroles containing alkyl substituents (Et, *n*-Pr, *n*-Bu) at the positions 4 and 5 of the azole ring, the expected cyclodimerization products were not detected at all. Also, this reaction did not proceed when acyl substituents were replaced by the ethoxycarbonyl group. Indeed, 2-ethoxycarbonylethynyltetrahydroindole did not form the expected dimers under standard conditions.

**Scheme 6.** Scope of 1-methylimidazole-catalyzed cyclodimerization of acylethynylpyrroles.

Nevertheless, the discovered cyclodimerization has a good chance of becoming more attractive for organic synthesis, provided that the reaction conditions will be optimized for each particular ethynylpyrrole. For example, the yield of dipyrrolopyrazine **2e** increased by 1.6 times (from 25 to 41%), when the corresponding ethynylpyrrole **1e** was dimerized in a two-phase phenanthridine/KOH/water system. Noteworthy, these conditions happened to be specific for this pyrrole **1e**, thus confirming the necessity to optimize the conditions for each case.

## 3. Discussion

The structure of compounds **2a–i** is easy recognizable due to the characteristic C2–C5 carbon signals of the pyrrole ring [30,31] combined with the C6–C8 carbon signals of the

olefin moiety and the carbonyl group (136.3, 120.5, 113.7, 120.9 and 123.6, 105.4, 188.5 ppm, respectively, for compound **2a** as instance, Figure 2).

**Figure 2.** Atom numbering and NOESY cross-peaks of the *E,E*- and *E,Z*-isomers of the compound **2a**.

*E,E*- and *E,Z*-configuration of product isomers was assigned according to 2D NOESY spectrum on the example of compound **2a**. The cross-peak between the H5 (7.45 ppm) and H7 (6.92 ppm) signals is observed in the $^1$H-$^1$H NOESY spectrum (see Supplementary Materials, Figure S3) of the *E,E*-isomer. In the *E,Z*-isomer, the cross-peak between the H5′ (7.45 ppm) and H7 (6.88 ppm) signals belongs to the moiety with the *E*-configuration, while the cross-peak between the H3′ (6.94 ppm) and H7′ (6.85 ppm) signals is attributed to the moiety with the *Z*-configuration (Figure 2).

In the *E,E*-isomer of compounds **2a–i**, both the **A** and **B** pyrrole rings (see Figure 2), both olefin fragments and carbonyl groups, as well as the R$^1$ and R$^2$ substituents in the **A** and **B** counterparts of the molecule, are equivalent due to the symmetry of the molecule. For this reason, one can observe the coincidence of the signals in both the $^1$H and $^{13}$C NMR spectra for the **A** and **B** counterparts of molecules **2a–i**. The symmetry of the considered molecules is broken upon passing from the *E,E*-isomer to the *E,Z*- one. It results in the appearance of two nonequivalent sets of signals in the $^1$H and $^{13}$C NMR spectra for the **A** and **B** pyrrole rings, as well as the corresponding olefin fragments, carbonyl groups, and the R$^1$ and R$^2$ substituents in the case of the *E,Z*-isomer. Therefore, the C2-C5 signals appear at 135.8, 119.7, 112.4, and 120.9 ppm, respectively, for the **A** pyrrole ring of compound **2a** while the C2′-C5′ signals can be found at 135.0, 114.0, 111.6, and 121.4 ppm, respectively, for the **B** pyrrole ring of the same compound. A similar difference between the **A** and **B** counterparts of molecule **2a** takes place for the C6-C8 signals of the olefin moiety and the carbonyl group, as well as the carbon signals of the phenyl ring as the R$^2$ substituent. At the same time, it should be noted that the exhaustive identification of signals in the $^1$H NMR spectrum of *E,Z*-isomers is difficult due to the partial overlap of the **A** and **B** counterpart signals of the compounds under discussion. Compounds **2a–c** with R$^1$ = H exist as a mixture of the *E,E*- and *E,Z*-isomers, the former being predominant (70–80% of the population). However, steric hindrance from the R$^1$ substituent (CH$_3$, (CH$_2$)$_4$, Ph, 4-FC$_6$H$_4$, 4-ClC$_6$H$_4$ or 2-thienyl) in compounds **2d–i** leads to the fact that only the *E,E*-isomer is realized in this case.

The *E,Z*-isomer of obtained products is a minor one, which suggests that the *Z*-arrangement of the carbonyl group with respect to the pyrrole ring is thermodynamically unfavorable, expectedly due to electronic repulsion between the carbonyl group and heterocyclic core. This is the most likely reason why the *Z,Z*-isomer is not formed.

Due to a lower content of the *E,Z*-isomers (20–30%) and close R-factors of both *E,E*- and *E,Z*-isomers, it was too laborious to separate them. In cases when the 5 position in pyrrole ring of the starting compounds is substituted, only *E,E*-isomers (ca. 100% selectivity) were synthesized and isolated.

A possible mechanism of dipyrrolopyrazines **2** formation can be rationalized as follows (Scheme 7). The lone electron pair of the "pyridine" nitrogen of the imidazole ring attacks the electrophilic C≡C triple bond of acetylenes **1** to generate 1,3-dipolar intermediate **A**, which, after intramolecular neutralization of the vinyl carbanionic center with a proton of the pyrrole moiety, forms 1,4-dipole **B** with a pyrrole nitrogen-centered anion. The nucleophilic attack of the latter on the electron-deficient triple bond of the second molecule of acetylenes **1** leads to the zwitterionic intermediate **C** with other vinyl carbanionic center. Undergoing a transformation similar to the **A→B** transformation, next pyrrole-centered dipole **D** is generated, which attacks the carbon of imidazolium site in a nucleophilic mode to release neutral imidazole, thereby completing the formation of pyrazine **2**.

**Scheme 7.** Possible mechanism of dipyrrolopyrazine **2a** formation via 1,3-dipole intermediates **A**.

The key step of the above mechanism, the formation of 1,3-dipole intermediate **A**, is the same as that of [3+2]-cycloaddition of acylethynylpyrroles to 1-pyrrolines [28]; however, it does not lead to similar prolongation, probably because of lower positive charge in the position 2 of the imidazole ring due to its distribution over the whole aromatic system.

The cyclodimerization catalyzed by other bases is less efficient (Tables 1 and 2) and likely proceeds via the following steps (Scheme 8). Pyrrolate anions **E**, produced by abstraction of a proton under the action of a base from the starting pyrroles, add as nucleophiles to the triple C≡C bond of the second ethynylpyrrole molecule **1** to give carbanions **F**, which are neutralized with the NH pyrrole proton. Nucleophilic intramolecular addition of the nitrogen-centered anion **G** to the triple bond and subsequent neutralization with a proton of another NH moiety affords dipyrrolopyrazine **2**.

We have performed [1]H NMR monitoring of the reaction of benzoylethynylpyrrole **1a** catalyzed by 1 equivalent of 1-methylimidazole (40 °C) in deuterated acetonitrile (CD₃CN) (see Supplementary Materials). The spectrum show the appearance (in 2 h after reaction start, spectrum 2, Figure S21) and slow increase of intensity (6 h, 96 h, 144 h, spectra 3–5, Figure S21) of the signals at 6.49, 7.74 and 8.04 ppm, which are attributed to the signals of the expected dimer, both opened and cyclic. The individual signals of intermediates and the product are most likely overlapped, since the structural fragments are almost identical. Moreover, it is probable that they do not appear at all owing to the high reactivity of these intermediates.

**Scheme 8.** Possible mechanism of dipyrrolopyrazine **2a** formation under the action of 1-methylimidazole as base.

The modest yields of the products originate from oligomerization of the intermediate dimer in a head-to-tail manner to give trimers, tetramers, including microcyclic ones and so on. In the case of benzoylethynylpyrrole **1a**, the oligomer represents a black-brown powder with wide ranges of m.p. (233–260 °C). Its $^1$H NMR spectrum consists of unresolved signals at 0.88, 1.24, 2.14 and 6.02–8.10 ppm. Elemental analysis of this oligomer (C, 79.90, H 4.28, N 6.94) is close to that of the initial ethynylpyrrole **1a** (C, 79.98, H 4.65, N 7.17).

## 4. Materials and Methods

### 4.1. General Considerations

NMR spectra were recorded on a Bruker DPX-400 spectrometer (Bruker, Billerica, MA, USA) (400.1 MHz for $^1$H, 100.6 MHz for $^{13}$C and 376.5 MHz for $^{19}$F) in CDCl$_3$. The internal standards were HMDS (for $^1$H), the residual solvents signal (for $^{13}$C) and CFCl$_3$ (for $^{19}$F). Coupling constants (*J*) were measured from one-dimensional spectra, and multiplicities were abbreviated as follows: s (singlet), d (doublet), dd (doublet of doublets), q (quartet), t (triplet), m (multiplet). IR spectra were recorded on a two-beam Bruker Vertex 70 spectrometer (Bruker, USA), in a microlayer, from chloroform. Mass spectra were recorded on an Agilent 6210 HRMS-TOF-ESI mass spectrometer (Agilent, Santa Clara, CA, USA) with electrostatic sputtering and registration of positive ions. Sample solvent was MeCN, with the addition of 0.1% heptafluorobutanoic acid and of a calibration mixture for the mass spectrometer. Melting points (uncorrected) were measured on a Kofler micro hot-stage apparatus (Wagner & Munz GmbH, München, Germany). 1-Methylimidazole was a commercial reagent. Acylethynylpyrroles **1** were obtained according to methods set out in [32]. Column and thin-layer chromatography for isolation and purification of compounds **2** were carried out on silica gel (0.06–0.2 mm) with chloroform/ethanol (20:1) mixture as eluent.

### 4.2. General Procedure for the Synthesis of Pyrazines 2

A mixture of acylethynylpyrrole **1** (0.5 mmol) and 1-methylimidazole (41 mg, 0.5 mmol) was stirred at 40–45 °C on a magnetic stirrer in an oil bath for an appropriate time. After the reaction stopped (control by IR spectroscopy by disappearance of the absorption band of the C≡C bond of **1** at 2155–2182 cm$^{-1}$), it was cooled to room temperature and passed through a chromatography column affording to dipyrrolo[1,2-*a*:1′,2′-*d*]pyrazine **2**.

## 5. Conclusions

The unprecedented 1-methylimidazole-catalyzed cyclodimerization of available acylethynylpyrroles, proceeding under mild conditions to afford pharmaceutically promis-

ing functionalized dipyrrolopyrazines of a novel structure, has been discovered. The discovered reaction contributes to the self-organization of pyrroles in the presence of organic catalysts. Despite the modest yields, the synthesized bis(acylmethylidene)dipyrrolo[1,2-*a*:1′,2′-*d*]pyrazines are of certain value as precursors for drug design and development of hi-tech materials (OLEDs and other electronic devices).

**Supplementary Materials:** The following supporting information can be downloaded at: https://www.mdpi.com/article/10.3390/catal12121604/s1. Experimental procedures, detailed characterization data and NMR spectra of all new compounds. Figure S1. $^1$H NMR spectrum of compound **2a**, Figure S2. $^{13}$C NMR spectrum of compound **2a**, Figure S3. 2D NMR NOESY spectrum of compound **2a**, Figure S4. $^1$H NMR spectrum of compound **2b**, Figure S5. $^{13}$C NMR spectrum of compound **2b**, Figure S6. $^1$H NMR spectrum of compound **2c**, Figure S7. $^{13}$C NMR spectrum of compound **2c**, Figure S8. $^1$H NMR spectrum of compound **2d**, Figure S9. $^{13}$C NMR spectrum of compound **2d**, Figure S10. $^1$H NMR spectrum of compound **2e**, Figure S11. $^{13}$C NMR spectrum of compound **2e**, Figure S12. $^1$H NMR spectrum of compound **2f**, Figure S13. $^{13}$C NMR spectrum of compound **2f**, Figure S14. $^1$H NMR spectrum of compound **2g**, Figure S15. $^{13}$C NMR spectrum of compound **2g**, Figure S16. $^{19}$F NMR spectrum of compound **2g**, Figure S17. $^1$H NMR spectrum of compound **2h**, Figure S18. $^{13}$C NMR spectrum of compound **2h**, Figure S19. $^1$H NMR spectrum of compound **2i**, Figure S20. $^{13}$C NMR spectrum of compound **2i**, Figure S21. $^1$H NMR monitoring of the reaction mixture between benzoylethynylpyrrole **1a** and 1-methylimidazole (1:1, 40 °C).

**Author Contributions:** K.V.B. and L.P.N. designed the experiments of the project; V.S.G. and D.N.T. performed the experiments; K.V.B. and L.A.O. drafted this manuscript; A.V.A. made assignments of NMR signals and established the structure; L.N.S. and B.A.T. proofread the manuscript and supervised the studies. All authors have read and agreed to the published version of the manuscript.

**Funding:** This work was financially supported by a grant from the Russian Science Foundation (project no. 21-73-10134).

**Acknowledgments:** The authors thank the Baikal Analytical Centre of collective use and Shared Research Facilities for Physical and Chemical Ultramicroanalysis, Limnological Institute, SB RAS (HRMS-TOF Spectra) for the equipment.

**Conflicts of Interest:** The authors declare no conflict of interest.

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
