# Peer review of "1-Methylimidazole as an Organic Catalyst for [3+3]-Cyclodimerization of Acylethynylpyrroles to Bis(acylmethylidene)dipyrrolo[1,2-a:1′,2′-d]pyrazines"

_catalysts, doi:10.3390/catal12121604_

Round 1
Reviewer 1 Report
In this work, A. Trofimov and co-workers describe an easy cyclodimerization in presence of 1-methylimidazole affording interesting dipyrrolo pyrazines.
Although the synthetic procedure uses mild conditions, the scope is very limited and the yields are very modest. I suggest to add new examples in the scope with higher yields. In addition, the comparison of cyclodimerization yields in presence of organic and inorganic basic catalysts (table 1 and 2) raises question on the validity of the proposed synthetic method. In order to clarify the mechanistic hypothesis (scheme 5 and 6), I suggest to perform an in situ spectroscopic monitoring of the process or alternatively planning some experiments devoted to detect diagnostic intermediates.
In the optimization of reaction conditions, what happens when you use protic solvents?
If the best yield is 51%, what about the side products?
In no case the mixture of diastereomers has been separated. Why?
In the supporting information, 13C spectra of compounds 2e and 2c show a not optimal noise/signal ratio. Please, I suggest to put new spectra for this compounds.
This work needs major revision in order to gain publication.
Author Response
Reviewer 1 wrote: “1. I suggest to add new examples in the scope with higher yields.”
Answer: To meet the referee request, we have extended the reaction scope over three compounds, namely, benzoylethynyl-(5-(4-fluorophenyl))pyrrole, benzoylethynyl-(5-(4-chlorophenyl))pyrrole and 2-(thenoylethynyl)-4,5,6,7-tetrahydroindole. However, the yields of the target products turned out to be not higher than those obtained with other studied compounds: 21%, 18% and 14%, respectively. It follows that the modest yields of functionalized dipyrrolopyrazine are immanent feature of the reaction under question: competition between cyclodimerization and oligomerization process. This, in degree, obvious drawback of the reaction is compensated by complexity and sophisticated structure of the compounds, which so far could not be obtained by any other method. It is clear that special development of the strategy for their synthesis would require elaboration of many steps that would lead at best to also not too high yields.
The additional results obtained for the three above acylethynylpyrroles are included in Scheme 4 and Supplementary materials.
Reviewer 1 wrote: “2. In addition, the comparison of cyclodimerization yields in presence of organic and inorganic basic catalysts (table 1 and 2) raises question on the validity of the proposed synthetic method. In order to clarify the mechanistic hypothesis (scheme 5 and 6), I suggest to perform an in situ spectroscopic monitoring of the process or alternatively planning some experiments devoted to detect diagnostic intermediates.”
Answer: To follow the reviewer suggestion, we have performed 1H NMR monitoring of the reaction of benzoylethynylpyrrole 1a catalyzed by 1 equivalent of 1-methylimidazole (40 oC) in deuterated acetonitrile (CD3CN) (see Supplementary materials). We observed appearance (in 2 h after reaction start, spectrum 2) and slow increase of intensity (6 h, 96 h, 144 h, spectra 3-5) of the signals at 6.49, 7.74 and 8.04 ppm, which are attributed to the signals of the expected dimer, both opened and cyclic one. The individual signals of intermediates and the product, most are likely overlapped, since the structural fragments are almost identical. Also it is probable that they do not appear at all owing to the high reactivity of these intermediates. Consequently, we are not sure that NMR monitoring could help to distinguish or anyhow detect the reaction intermediates.
The corresponding spectra have been placed to Supplementary materials and the following sentences were included to the manuscript.
“We have performed 1H NMR monitoring of the reaction of benzoylethynylpyrrole 1a catalyzed by 1 equivalent of 1-methylimidazole (40 oC) in deuterated acetonitrile (CD3CN) (see Supplementary materials). The spectrum show the appearance (in 2 h after reaction start, spectrum 2) and slow increase of intensity (6 h, 96 h, 144 h, spectra 3-5) of the signals at 6.49, 7.74 and 8.04 ppm, which are attributed to the signals of the expected dimer, both opened and cyclic one. The individual signals of intermediates and the product, most are likely overlapped, since the structural fragments are almost identical. Also it is probable that they do not appear at all owing to the high reactivity of these intermediates.”
Reviewer 1 wrote: “3. In the optimization of reaction conditions, what happens when you use protic solvents?”
Answer: In the presence of bases (amines, alkaline metal hydroxides or alcoxides). All protogenic solvents including alcohols easily react with electron-deficient triple bond to give the corresponding adducts (Synthesis, 2010, N. 14, 2468-2474; Synthesis 2017, 49, 4065-4081). Therefore, it is of no reason to use these solvents in the reaction studied. Nevertheless, as additional experiment showed (Table 2, entry 5) in water only traces of expected product was detected in the reaction mixture.
Reviewer 1 wrote: “4. If the best yield is 51%, what about the side products?”
Answer: The second part of the loaded ethynylpyrroles is oligomerized obviously, in intramolecular manner, and undergoes head-to-tail addition to produce open-chain and, probably, macrocyclic trimers, tetramers and other oligomers. For example, in the case of benzoylethynylpyrrole 1a, the oligomer (the side product) represents a black-brown powder with wide ranges of m.p. (233-260 oC). Its 1H NMR spectrum consists of unresolved signals at 0.88, 1.24, 2.14 and 6.02-8.10 ppm. Elemental analysis of this oligomer (C, 79.90, H 4.28, N 6.94) is close to that of the initial ethynylpyrrole 1a (C, 79.98, H 4.65, N 7.17).
The following sentences were added to the text: “The modest yields of the products are originated from oligomerization of the intermediate dimer in a head-to-tail manner to give trimers, tetramers, including microcyclic ones and so one. In the case of benzoylethynylpyrrole 1a, the oligomer represents a black-brown powder with wide ranges of m.p. (233-260 oC). Its 1H NMR spectrum consists of unresolved signals at 0.88, 1.24, 2.14 and 6.02-8.10 ppm. Elemental analysis of this oligomer (C, 79.90, H 4.28, N 6.94) is close to that of the initial ethynylpyrrole 1a (C, 79.98, H 4.65, N 7.17).”
Reviewer 1 wrote: “5. In no case the mixture of diastereomers has been separated. Why?”
Answer: Due to a lower content of the E,Z-isomers (20-30%) and close R-factors of both E,E- and E,Z-isomers and, it was too laborious to separate them. In cases, when the 5 position in pyrrole ring of the starting compounds is substituted, only E,E-isomers (ca. 100% selectivity) were synthesized and isolated.
These sentences were included to the text.
Reviewer 1 wrote: “6. In the supporting information, 13C spectra of compounds 2e and 2c show a not optimal noise/signal ratio. Please, I suggest to put new spectra for this compounds”
Answer: Done as recommended.
Reviewer 2 Report
Trofimov and co-workers reported an interesting 1-methylimidazole-catalyzed cyclodimerization of acylethynylpyrroles under mild conditions to afford pharmaceutically useful dipyrrolopyrazines. The related findings are valuble to the corresponding readers. I recommend to accept this manuscript after minor revision.
1. I suggests using "Тemp., оС" and "Time, h" to substitute "Т, оС" and "T, h" in Table 1 and Table 2, avoiding the confusion.
2. How to determine the E and Z construction of products? It should be clearly described in the manuscript at appropriate position.
3. For products , only E,E- and E,Z- structures were observed, and no Z,Z- was obtained, why? It should also be described in the manuscript.
Author Response
Reviewer 2 wrote: “2. How to determine the E and Z construction of products? It should be clearly described in the manuscript at appropriate position.
Answer: E,E- and E,Z-configurations of the product isomer were determined according to 2D NOESY spectrum of compound 2a. In the spectrum of the E,Z-isomer, the cross-peak between H5’ (7.45 ppm) and H7 (6.88 ppm) signals belongs to the moiety with the E-configuration, while the cross-peak between H3’ (6.94 ppm) and H7’ (6.85 ppm) signals is attributed to the moiety with the Z-configuration (see Supplementary materials).
The following sentences were added to the text of manuscript:
The Figure 2 was also added to the manuscript.
“E,E- and E,Z-configurations of the product isomers were assigned according to 2D NOESY spectra (see Supplementary materials). In the 1H-1H NOESY spectrum of compound 2a, the cross-peak between H5 (7.45 ppm) and H7 (6.92 ppm) signals was observed (see Supplementary materials), which belongs to the E,E-isomer. In the E,Z-isomer, the cross-peak between the H5’ (7.45 ppm) and H7 (6.88 ppm) signals belongs to the E-configuration, while the cross-peak between the H3’ (6.94 ppm) and H7’ (6.85 ppm) signals is attributed to the Z-configuration (Figure 2).”
Reviewer 2 wrote: “3. For products , only E,E- and E,Z- structures were observed, and no Z,Z- was obtained, why? It should also be described in the manuscript.”
Answer: The E,Z-isomer of the obtained products is a minor one, which suggests that the Z-arrangement of the carbonyl group with respect to the pyrrole ring is thermodynamically unfavorable, expectedly due to electronic repulsion between the carbonyl group and heterocyclic core. This is the most likely reason why the Z,Z-isomer is not formed.
This explanation was added to the text of manuscript.
Round 2
Reviewer 1 Report
The authors have replied to all reviewer's requests in satisfactory manner furnishing suitable explanations in the text and supporting information. Accordingly, the work is suitable for publication in Catalysts.